# Point or Line? Using Line-based Representation for Panoptic Symbol Spotting in CAD Drawings

Xingguang Wei[1,2*]  Haomin Wang[3,2*]  Shenglong Ye[2]
Ruifeng Luo[4,8]  Yanting Zhang[5]  Lixin Gu[2]  Jifeng Dai[2,6]
Yu Qiao[2]  Wenhai Wang[2,7]  Hongjie Zhang[2†]

[1]University of Science and Technology of China  [2]Shanghai AI Laboratory
[3]Shanghai Jiao Tong University  [4]Tongji University  [5]Donghua University
[6]Tsinghua University  [7]The Chinese University of Hong Kong
[8]Arcplus East China Architectural Design & Research Institute Co., Ltd.

The code is available at `https://github.com/WesKwong/VecFormer`

## Abstract

We study the task of panoptic symbol spotting, which involves identifying both individual instances of countable *things* and the semantic regions of uncountable *stuff* in computer-aided design (CAD) drawings composed of vector graphical primitives. Existing methods typically rely on image rasterization, graph construction, or point-based representation, but these approaches often suffer from high computational costs, limited generality, and loss of geometric structural information. In this paper, we propose *VecFormer*, a novel method that addresses these challenges through *line-based representation* of primitives. This design preserves the geometric continuity of the original primitive, enabling more accurate shape representation while maintaining a computation-friendly structure, making it well-suited for vector graphic understanding tasks. To further enhance prediction reliability, we introduce a *Branch Fusion Refinement* module that effectively integrates instance and semantic predictions, resolving their inconsistencies for more coherent panoptic outputs. Extensive experiments demonstrate that our method establishes a new state-of-the-art, achieving 91.1 PQ, with Stuff-PQ improved by 9.6 and 21.2 points over the second-best results under settings with and without prior information, respectively—highlighting the strong potential of line-based representation as a foundation for vector graphic understanding.

## 1 Introduction

Panoptic symbol spotting refers to the task of detecting and classifying all symbols within a CAD drawing, including both countable object instances (*e.g.*, windows, doors, furniture) and uncountable stuff regions (*e.g.*, walls, railings) [1, 2, 3]. This capability is crucial in CAD-based applications, serving as a foundation for automated design review and for generating 3D Building Information Models (BIM). However, spotting each symbol, which typically comprises a group of graphical primitives, remains highly challenging due to factors such as occlusion, clutter, appearance variations, and severe class imbalance across different symbol categories.

Earlier approaches to this problem either rasterize CAD drawings and apply image-based detection or segmentation methods [1, 4], or directly construct graph representations of CAD drawings and leverage GNN-based techniques [5, 6, 7]. However, both paradigms incur substantial computational costs, particularly when applied to large-scale CAD drawings. To better handle primitive-level

---

*Equal Contribution

†Corresponding author: nju.zhanghongjie@gmail.com

39th Conference on Neural Information Processing Systems (NeurIPS 2025).

data, recent methods treat CAD drawings as sets of points corresponding to graphical primitives and leverage point cloud analysis for symbol spotting. For example, SymPoint [8] represents each primitive as a point with handcrafted features, encoding attributes such as primitive type and length. However, this manually defined representation is restricted to four predefined primitive types (line, arc, circle, and ellipse) and struggles to accommodate the more complex and diverse shapes frequently encountered in real-world CAD drawings. In contrast, the recent CADSpotting [9] forgoes explicit primitive types by densely sampling points along each primitive and representing each point using only its coordinate and color. Although this design eliminates reliance on primitive types, it lacks geometric structure and primitive-level awareness, which may hinder the model's ability to delineate symbol boundaries, resolve overlapping symbols, and capture structural configurations essential for accurate symbol spotting.

In this work, we propose *VecFormer*, a Transformer-based [10] model built on a *line-based representation* that serves as an expressive and type-agnostic formulation for vector graphical primitives. It employs line sampling to generate a sequence of line segments along each primitive, with each line represented by its intrinsic geometric attributes and associated primitive-level statistics, forming a compact and informative feature set. Figure 1 illustrates a visual comparison of different primitive representations. SymPoint [8] encodes each primitive as a single point, which is too coarse to capture the fine-grained structures, especially for long primitives commonly found in stuff regions, leading to degraded performance. To ensure a fair comparison, we adopt the same sampling density across sampling-based methods. As shown in Figure 1, unlike CADSpotting [9] which suffers from blurred symbol boundaries, our line-based VecFormer yields results with clearer structure and better alignment to ground-truth, demonstrating higher geometric and structural fidelity. This more compact yet expressive representation is also better suited for Transformer-based architecture, which is sensitive to input sequence length. Further discussion on sequence length across different representations is detailed in Appendix C.

Additionally, inspired by OneFormer3D [11], we adopt a dual-branch Transformer decoder to guide the representation learning of vector graphical primitives, leveraging its strong multi-tasking capability to jointly model instance- and semantic-level information. To produce a more coherent panoptic output, we further propose a lightweight, training-free post-processing module, termed *Branch Fusion Refinement* (BFR), which combines predictions from the instance and semantic branches through confidence-based fusion. This refinement enhances label consistency, mitigates mask fragmentation, and improves the overall coherence of panoptic symbol predictions.

To summarize, our main contributions are:

(1) We introduce *VecFormer*, a novel approach that leverages a type-agnostic and expressive *line-based representation* of vector graphcal primitives, instead of traditional point-based methods, leading to more accurate and efficient panoptic symbol spotting.

(2) We propose a *Branch Fusion Refinement* (BFR) module that effectively integrates instance and semantic predictions via confidence-based fusion, resolving their inconsistencies for more coherent panoptic outputs, yielding a performance gain of approximately 2 points in panoptic quality (PQ) on the FloorPlanCAD [1] dataset.

(3) We conduct extensive experiments on the FloorPlanCAD [1] dataset, where our *VecFormer* achieves a PQ of 91.1, setting a new state-of-the-art in the panoptic symbol spotting task. Notably, it improves Stuff-PQ by 9.6 and 21.2 points over the second-best results under settings with and without prior information, respectively, underscoring its superior performance and robustness in real-world CAD applications.

## 2   Related Work

### 2.1   Panoptic Image Segmentation

Panoptic segmentation [12] aims to unify semantic [13, 14, 15, 16, 17] and instance segmentation [18, 19, 20, 21] by assigning each pixel both a class label and an instance ID, effectively covering both *things* (countable objects) and *stuff* (amorphous regions). Early approaches predominantly relied on CNN-based architectures [22, 23, 24, 25], which, while effective, often required separate branches for different segmentation tasks. Recent advancements have seen a shift towards Transformer-based models, which offer unified architectures for various segmentation tasks. Notably, Mask2Former [26]

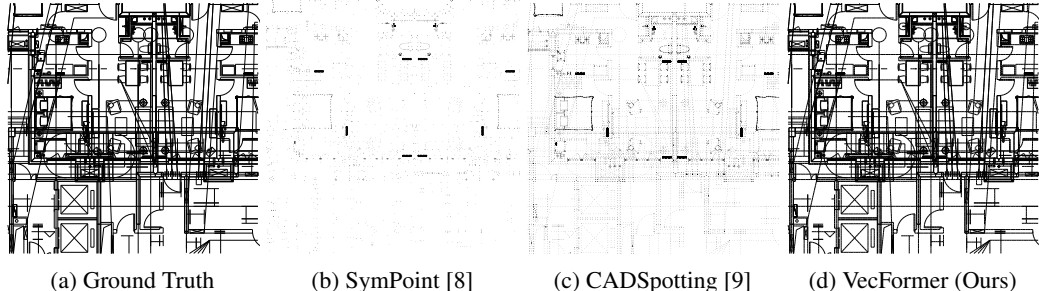

| (a) Ground Truth | (b) SymPoint [8] | (c) CADSpotting [9] | (d) VecFormer (Ours) |

Figure 1: **Visualization of primitive representations.** Compared to the blurry visual representations of point-based methods (b, c), our line-based approach (d) more closely reflects the ground truth drawing (a). As the data is in vector format, please feel free to zoom in to observe finer differences. Additional comparisons are provided in Appendix A and Appendix B.

unifies panoptic, instance, and semantic segmentation using masked attention. SegFormer [17] improves efficiency with hierarchical encoders and lightweight decoders. OneFormer [20] further introduces task-conditioned training to jointly handle multiple segmentation tasks. Despite these successes in raster image domains, pixel-centric segmentation models face challenges when applied to vector graphics tasks, such as Panoptic Symbol Spotting in CAD drawings. Their reliance on dense pixel grids overlooks the inherent structure of vector primitives, making it difficult to capture precise geometric relationships, maintain topological consistency, and resolve overlapping symbols. These limitations hinder performance in structured, symbol-rich vector environments.

## 2.2 Panoptic Symbol Spotting

The panoptic symbol spotting task, first introduced in [1], aims to simultaneously detect and classify architectural symbols (*e.g.*, doors, windows, walls) in floor plan computer-aided design (CAD) drawings. While earlier approaches [2] primarily addressed instances of countable *things* (*e.g.*, windows, doors, tables), Fan *et al.* [1], inspired by [12], extended the task to include semantic regions of uncountable *stuff* (*e.g.*, wall, railing). To support this task, they introduced the FloorPlanCAD benchmark and proposed PanCADNet as a baseline, which combines Faster R-CNN [27] for detecting countable *things* with Graph Convolutional Networks [28] for segmenting uncountable *stuff*. Subsequently, Fan *et al.* [4] proposed CADTransformer, utilizing HRNetV2-W48 [29] and Vision Transformers [30] for primitive tokenization and embedding aggregation. Zheng *et al.* [6] adopted graph-based representations with Graph Attention Networks [31] for instance- and semantic-level predictions. Liu *et al.* [8] introduced SymPoint, exploring point-based representations with handcrafted features, later enhanced by SymPoint-V2 [32] through layer feature encoding and position-guided training. Recently, CADSpotting [9] densely samples points along primitives to generate dense point data for feature extraction and employs Sliding Window Aggregation for efficient panoptic segmentation of large-scale CAD drawings. Although point-based representations are widely adopted in existing state-of-the-art methods [8, 32, 9], they exhibit notable limitations in complex and densely annotated CAD drawings, including redundant sampling, loss of geometric continuity, and reduced ability to distinguish adjacent or overlapping symbols, as shown in Figure 1.

## 3 Method

In this section, we first describe how heterogeneous vector graphic primitives are converted into a unified line-based representation. We then present the panoptic symbol spotting framework built upon this representation. Finally, we introduce our post-processing optimization strategy, Branch Fusion Refinement. An overview of the entire pipeline is shown in Figure 2.

## 3.1 Line Sampling

Existing point-based representations [8, 32, 9] suffer from limited geometric continuity, structural expressiveness, and generality across diverse primitive types. To address these issues, we propose

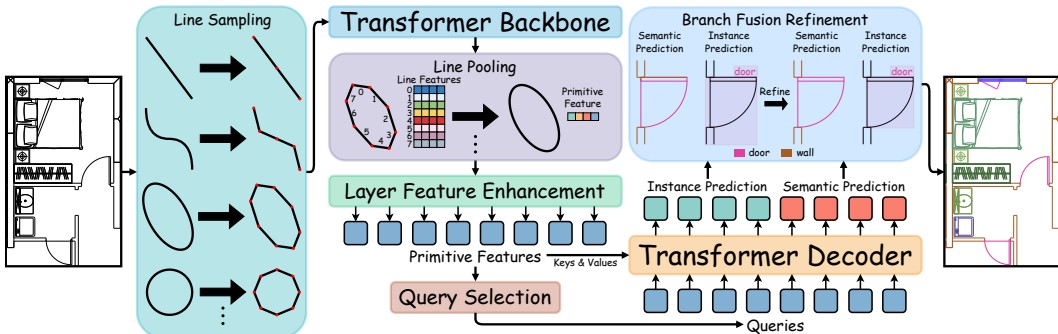

Figure 2: **Overview of VecFormer.** Given a CAD drawing, VecFormer first applies line sampling to build a line-based representation of primitives. A Transformer backbone is then used to extract line-level features, which are subsequently aggregated into primitive-level features. Next, these primitive-level features are enhanced by a Layer Feature Enhancement module and fed into a Transformer decoder for joint instance and semantic prediction. Finally, a Branch Fusion Refinement module integrates both branches to produce the final panoptic symbol spotting result.

*Line Sampling*, a line-based approximation that encodes primitives as sequences of line segments, enabling unified and geometry-preserving modeling of heterogeneous vector graphics.

Specifically, given a vector primitive with a unique identifier $j$, we first convert it into a vector path $\gamma_j(t) : [0, 1] \to \mathbb{R}^2$. Then, we perform uniform or dynamic path sampling over its parameter interval to generate a sequence of sampled points $\mathcal{P}_j = \{\mathbf{p}_i = \gamma_j(t_i) \mid i = 1, \ldots, K\}$. Here, $0 = t_1 < t_2 < \cdots < t_K = 1$, where the number of samples $K$ and the sampling parameters $t_i$ can be dynamically adjusted based on geometric features such as the length and curvature of the primitive.

For simplicity, we adopt a uniform sampling strategy defined as: $t_i = \frac{i-1}{K-1}$, and use a hyperparameter called the *sampling ratio* to control the number of samples $K$. Specifically, for line primitives, we initially set $K = 2$; for all other types of primitives, we initially set $K = 9$. Given a sampling ratio $\alpha_{\text{sample}}$, we constrain the maximum allowable distance between adjacent sample points to be no greater than $\alpha_{\text{sample}} \cdot \min(\text{width}, \text{height})$, where *width* and *height* denote the dimensions of the CAD drawing. If this condition is violated, we iteratively increase the number of samples by setting $K \leftarrow K + 1$ until the constraint is satisfied.

Next, adjacent sampling points are pairwise connected to construct a sequence of line segments:

$$\mathcal{L}_j = \{\mathbf{l}_i = (\mathbf{p}^s, \mathbf{p}^e) \mid \mathbf{p}^s = \mathbf{p}_i, \mathbf{p}^e = \mathbf{p}_{i+1}, i = 1, \ldots, K-1\}, \tag{1}$$

which approximates the geometric features of the original primitive.

## 3.2 Panoptic Symbol Spotting via Line-based Representation

The process of panoptic symbol spotting via the line-based representation consists of three main stages: first, using a backbone to extract line-level features; second, pooling the line-level features into primitive-level features; and third, utilizing a 6-layer Transformer decoder to generate instance proposals and semantic predictions.

### 3.2.1 Backbone

We choose Point Transformer V3 (PTv3) [33] as our backbone for feature extraction due to its excellent performance in handling unordered data with irregular spatial distributions.

Given a sampled line $\mathbf{l}_i$, with its starting point $\mathbf{p}^s = (x_1, y_1)$ and endpoint $\mathbf{p}^e = (x_2, y_2)$, the primitive ID $j$ indicates the primitive to which the line segment belongs, and the layer ID $k$ indicates the layer on the CAD drawing where the primitive is located. we will now describe how to convert it into the position vector $\mathbf{coord}_i \in \mathbb{R}^3$ and the corresponding feature vector $\mathbf{feat}_i \in \mathbb{R}^C$ ($C$ is the dimensionality of the feature vector) suitable for input to the PTv3 backbone.

**Normalization.** The initial step involves the normalization of the raw line features to a standardized range of $[-0.5, 0.5]$.

For the starting point $\mathbf{p}^s = (x_1, y_1)$, the normalization is performed as follows:

$$\mathbf{p}^s = (x_1, y_1) = \left( \frac{x_1 - x_{\text{origin}}}{\text{width}} - 0.5, \frac{y_1 - y_{\text{origin}}}{\text{height}} - 0.5 \right). \tag{2}$$

In this formulation, the coordinates $(x_{\text{origin}}, y_{\text{origin}})$ denote the origin of the coordinate system employed in the CAD drawing. The terms *width* and *height* denote the dimensions of the CAD drawing. The normalization for the endpoint $\mathbf{p}^e$ is achieved through an analogous transformation.

For the normalization of layer ID, let $k_{\text{min}}$ and $k_{\text{max}}$ represent the minimum and maximum layer ID values observed within the CAD drawing, respectively. The normalized layer ID $k$ is then calculated as:

$$k = \frac{k - k_{\text{min}}}{k_{\text{max}} - k_{\text{min}}} - 0.5. \tag{3}$$

**Line Position.** To simultaneously capture both the position information and the layer information, we use the midpoint of the line $(c_x, c_y)$ for the first two dimensions and the layer ID $k$ for the third dimension:

$$\mathbf{coord}_i = (x_i, y_i, z_i) = (c_x, c_y, \text{id}) = \left( \frac{x_1 + x_2}{2}, \frac{y_1 + y_2}{2}, k \right). \tag{4}$$

**Line Feature.** We set the dimensionality $C = 7$, and define the line feature $\mathbf{feat}_i \in \mathbb{R}^7$ as:

$$\mathbf{feat}_i = (l, d_x, d_y, c_x, c_y, c_x^p, c_y^p). \tag{5}$$

Here, $l = \sqrt{(x_2 - x_1)^2 + (y_2 - y_1)^2}$ represents the length of the line. The terms $d_x = (x_1 - x_2)/l$ and $d_y = (y_1 - y_2)/l$ denote the unit vectors for displacement in the $x$ and $y$ directions, respectively. The coordinates $(c_x, c_y)$ specify the midpoint of the line. These features are chosen because any point on the line can be expressed as: $(x, y) = (c_x + td_x, \ c_y + td_y), t \in [-\frac{l}{2}, \frac{l}{2}]$, which provides a parametric representation of the line segment based on its center and unit vector.

Furthermore, $(c_x^p, c_y^p)$ indicates the geometric centroid of the primitive. This centroid is determined by calculating the average of the midpoint coordinates from all lines sampled within the primitive $j$ to which the specific line belongs:

$$(c_x^p, c_y^p) = \left( \frac{\sum_{\mathbf{l}_i \in \mathcal{L}_j} c_x}{|\mathcal{L}_j|}, \frac{\sum_{\mathbf{l}_i \in \mathcal{L}_j} c_y}{|\mathcal{L}_j|} \right). \tag{6}$$

### 3.2.2 Line Pooling

To obtain primitive-level features, we apply *Line Pooling*, which combines max and average pooling over line-level features within each primitive, effectively preserving geometric information and enhancing feature richness.

For each primitive $j$, line features $\mathbf{f}_i \in \mathbb{R}^C$ from $\mathbf{l}_i \in \mathcal{L}_j$ are aggregated via both max and average pooling, whose results are summed to produce the final primitive feature $\mathbf{F}_j \in \mathbb{R}^C$:

$$\mathbf{F}_j = \mathbf{F}_j^{\text{max}} + \mathbf{F}_j^{\text{avg}} = \max_{\mathbf{l}_i \in \mathcal{L}_j} \mathbf{f}_i + \frac{1}{|\mathcal{L}_j|} \sum_{\mathbf{l}_i \in \mathcal{L}_j} \mathbf{f}_i. \tag{7}$$

### 3.2.3 Layer Feature Enhancement

Inspired by SymPoint-V2 [32], we adopt a *Layer Feature Enhancement* (LFE) module in our method. Specifically, we aggregate the features of primitives within the same layer using average pooling, max pooling, and attention pooling, and fuse the resulting layer-level context back into each primitive feature. This fusion enhances the model's ability to capture intra-layer contextual dependencies and improves the semantic discrimination of similar primitives.

### 3.2.4 Query Decoder

Motivated by OneFormer3D [11], we initialize the queries using a *Query Selection* strategy, which is widely adopted in state-of-the-art 2D object detection and instance segmentation methods [34, 35, 36].

Subsequently, a six-layer Transformer decoder performs self-attention on the queries and cross-attention with key-value pairs derived from primitive features. The decoder outputs are then passed to an *Instance Branch* for generating instance proposals and a *Semantic Branch* for producing semantic predictions.

**Query Selection.** With the primitive features $\mathcal{L} \in \mathbb{R}^{N \times C}$ derived from the previous stage, where $N$ denotes the number of primitives and $C$ is the dimensionality of each feature vector, the Query Selection strategy randomly selects a proportion $\alpha_{\text{select}} \in [0, 1]$ of the primitive features to initialize the queries $\mathcal{Q} \in \mathbb{R}^{M \times C}$, with $M = \alpha_{\text{select}} \cdot N$ representing the number of queries. Following the configuration in OneFormer3D [11], we set $\alpha_{\text{select}} = 0.5$ during training to reduce computational cost, which also serves as a form of data augmentation. During inference, we set $\alpha_{\text{select}} = 1.0$ in order to preserve the complete information of the CAD drawings.

**Instance Branch.** In this branch, each query embedding is mapped to a $K + 1$ dimensional space as class label logits, where $K$ denotes the number of classes and an extra $+1$ for the background predictions. Simultaneously, we use an `einsum` operation between the query embedding and the primitive features to generate the instance mask.

**Semantic Branch.** This branch aims to produce dense, per-primitive semantic predictions. We project the output queries from the decoder into a $K + 1$ dimensional space as semantic logits. The prediction for each query is assigned to the primitive that was selected to initialize the query during the Query Selection process, thereby providing semantic label of each primitive.

### 3.2.5 Loss Function

To jointly optimize instance and semantic predictions, we adopt a composite loss function:

$$L_{\text{total}} = \lambda_{\text{cls}} L_{\text{cls}} + \lambda_{\text{bce}} L_{\text{bce}} + \lambda_{\text{dice}} L_{\text{dice}} + \lambda_{\text{sem}} L_{\text{sem}}. \tag{8}$$

Here, $L_{\text{cls}}$ is a cross-entropy loss for instance classification, $L_{\text{bce}}$ and $L_{\text{dice}}$ [37, 38] are used for instance mask prediction to balance foreground-background accuracy and mask overlap, respectively. $L_{\text{sem}}$ denotes the cross-entropy loss for semantic segmentation. The weights $\lambda_{\text{cls}}, \lambda_{\text{bce}}, \lambda_{\text{dice}}, \lambda_{\text{sem}}$ control the influence of each term.

## 3.3 Branch Fusion Refinement

To effectively integrate information from both the Semantic Branch and the Instance Branch, we propose a post-processing strategy named *Branch Fusion Refinement* (BFR). This method consists of three steps: *Overriding*, *Voting*, and *Remasking*.

**Overriding.** This step is primarily designed to resolve conflicts between instance predictions and semantic predictions at the per-primitive level. Given a primitive $p_i$, the semantic branch outputs a semantic label $l_{\text{sem}}(p_i) \in \{1, \ldots, K + 1\}$ and a corresponding confidence score $s_{\text{sem}}(p_i) \in [0, 1]$. Meanwhile, if $p_i$ is assigned to $N$ instance proposals, each such proposal provides an instance label $l_{\text{inst}}^j \in \{1, \ldots, K + 1\}$ and an associated confidence score $s_{\text{inst}}^j \in [0, 1]$, where $j \in \{1, \ldots, N\}$ indexes the proposals that include $p_i$.

To resolve the conflict, we compare the semantic and instance confidence scores. If the highest instance score for $p_i$ is greater than the semantic score, i.e., $\max_j s_{\text{inst}}^j(p_i) > s_{\text{sem}}(p_i)$, then the semantic prediction for $p_i$ is overridden by the instance label and score of the highest-confidence proposal:

$$l_{\text{sem}}^{\text{refined}}(p_i) = l_{\text{inst}}^{j^*}(p_i), \quad s_{\text{sem}}^{\text{refined}}(p_i) = s_{\text{inst}}^{j^*}(p_i), \quad \text{where} j^* = \arg \max_{j \in \{1, \ldots, N\}} s_{\text{inst}}^j(p_i). \tag{9}$$

If no instance score exceeds the semantic score, the original semantic prediction is retained.

**Voting.** Given an instance proposal that contains $M$ primitives, its instance label is refined based on the most frequently occurring semantic class among those primitives. Formally, the instance label $l_{\text{inst}}$ for this proposal is refined as:

$$l_{\text{inst}} = \arg \max_{k \in \{1, \ldots, K\}} \sum_{i=1}^{M} \mathbb{I}\left(l_{\text{sem}}(p_i) = k\right), \tag{10}$$

where $\mathbb{I}(\cdot)$ is the indicator function that returns 1 if the condition is true and 0 otherwise. This majority voting strategy ensures that the instance label aligns with the dominant semantic context of its constituent primitives.

**Remasking.** For each primitive $p_i$, if it belongs to an instance mask $\mathcal{M}_{\text{inst}}$, but its semantic label $l_{\text{sem}}(p_i)$ disagrees with the instance's majority-voted label $l_{\text{inst}}$, it is removed from the mask:

$$p_i \in \mathcal{M}_{\text{inst}} \quad \text{and} \quad l_{\text{sem}}(p_i) \neq l_{\text{inst}} \quad \Rightarrow \quad p_i \notin \mathcal{M}_{\text{inst}}. \tag{11}$$

This operation effectively eliminates label contamination in the mask caused by prediction inconsistencies, thereby improving the purity and semantic consistency of the instance segmentation results.

## 4 Experiments

### 4.1 Dataset and Metrics

FloorPlanCAD [1] dataset consists of 11,602 diverse CAD drawings of various floor plans, each annotated with fine-grained semantic and instance labels. We follow the official data split, which includes 6,965 samples for training, 810 for validation, and 3,827 for testing. The annotations cover 30 thing classes and 5 stuff classes.

Following [1, 4], we use the Panoptic Quality (PQ) defined on vector graphics as our main metric to evaluate the performance of panoptic symbol spotting. The Panoptic Quality (PQ) serves as a comprehensive metric that simultaneously evaluates the recognition correctness and segmentation accuracy of symbol-level predictions in vector graphics. A graphical primitive is denoted as $e = (l, z)$, where $l$ is the semantic label, $z$ is the instance index. A symbol is represented by a collection of primitives and is defined as $s = \{e_i \in J \mid l = l_i, z = z_i\}$, where J is a set of primitives. The metric is defined as:

$$PQ = \frac{|TP|}{|TP| + \frac{1}{2}|FP| + \frac{1}{2}|FN|} \times \frac{\sum_{(s_p, s_g) \in TP} \text{IoU}(s_p, s_g)}{|TP|} = \frac{\sum_{(s_p, s_g) \in TP} \text{IoU}(s_p, s_g)}{|TP| + \frac{1}{2}|FP| + \frac{1}{2}|FN|}. \tag{12}$$

Here, $s_p = (l_p, z_p)$ is the predicted symbol, and $s_g = (l_g, z_g)$ is the ground truth symbol. $|TP|$, $|FP|$, and $|FN|$ represent the number of true positives, false positives, and false negatives, respectively. A predicted symbol is matched to a ground truth symbol if and only if $l_p = l_g$ and $\text{IoU}(s_p, s_g) > 0.5$. The IoU between two symbols is defined as:

$$\text{IoU}(s_p, s_g) = \frac{\sum_{e_i \in s_p \cap s_g} \log(1 + L(e_i))}{\sum_{e_j \in s_p \cup s_g} \log(1 + L(e_j))}, \tag{13}$$

where $L(e)$ denotes the length of a geometric primitive $e$.

### 4.2 Implementation Details

During training, we adopt the AdamW [39] optimizer with a weight decay of 0.05. The initial learning rate is set to 0.0001, with a warm-up ratio of 0.05, followed by cosine decay applied over 20% of the total training epochs. The model is trained for 500 epochs with a batch size of 2 per GPU on 8 NVIDIA A100 GPUs. To improve model generalization, we apply several data augmentation strategies during training, including random horizontal and vertical flips with a probability of 0.5, random rotations, random scaling within the range [0.8, 1.2], and random translations up to 10% of the CAD drawing size along both axes. Furthermore, we empirically set the loss weight as $\lambda_{\text{cls}} : \lambda_{\text{bce}} : \lambda_{\text{dice}} : \lambda_{\text{sem}} = 2.5 : 5.0 : 5.0 : 5.0$.

### 4.3 Quantitative Evaluation

**Panoptic Symbol Spotting.** We compare our method with existing approaches on FloorPlanCAD [1] for panoptic symbol spotting, as shown in Table 1a. Our method achieves the highest Panoptic Quality (PQ) across *Total*, *Thing*, and *Stuff* categories, demonstrating superior and more balanced performance. Existing methods tend to perform better on *Thing* than *Stuff* categories, revealing an imbalance in recognition. For example, SymPoint [8] scores 84.1 in Thing-PQ but only 48.2 in

Table 1: Quantitative evaluation results

(a) Panoptic symbol spotting results on FloorPlanCAD [1] dataset. A dash (–) indicates that the method does not support this setting or that the result is not reported in the original paper.

| Method | w/o Prior | | | w/ Prior | | |
|---|---|---|---|---|---|---|
| | PQ | $PQ_{th}$ | $PQ_{st}$ | PQ | $PQ_{th}$ | $PQ_{st}$ |
| PanCADNet [1] | 59.5 | 65.6 | 58.7 | - | - | - |
| CADTransformer [4] | 68.9 | 78.5 | 58.6 | - | - | - |
| GAT-CADNet [6] | 73.7 | - | - | - | - | - |
| SymPoint [8] | 83.3 | 84.1 | 48.2 | - | - | - |
| SymPoint-V2 [32] | 83.2 | 85.8 | 49.3 | 90.1 | 90.8 | 80.8 |
| CADSpotting [9] | - | - | - | 88.9 | 89.7 | 80.6 |
| DPSS [40] | 86.2 | 88.0 | 64.7 | 89.5 | 90.4 | 79.7 |
| **VecFormer (Ours)** | **88.4** (+2.2) | **90.9** (+2.9) | **85.9** (+21.2) | **91.1** (+1.0) | **91.8** (+1.0) | **90.4** (+9.6) |

(b) Primitive-level semantic quality. wF1: length-weighted F1.

| Method | GAT-CADNet [6] | SymPoint [8] | SymPoint-V2 [32] | **VecFormer (Ours)** |
|---|---|---|---|---|
| F1 | 85.0 | 86.8 | 89.5 | **93.8** (+4.3) |
| wF1 | 82.3 | 85.5 | 88.3 | **92.2** (+3.9) |

Stuff-PQ. In contrast, our method achieves more balanced results and shows a marked advantage in the *Stuff* classes, in particular, surpassing the current state-of-the-art method, SymPoint-V2 [32], by 9.6 in Stuff-PQ.

To reflect real-world conditions where detailed annotations (such as layers) are often unavailable, we evaluate current mainstream methods without using prior information. As shown in Table 1a, existing state-of-the-art methods exhibit strong reliance on prior, particularly for *Stuff* categories. Specifically, SymPoint-V2 [32] and DPSS [40] suffer significant performance drops in Stuff-PQ when evaluated without prior, decreasing by 31.5 and 15 points, respectively. In contrast, our method VecFormer consistent performance across both settings by using primitive IDs instead of layer IDs as $z$-coordinate of position vector, i.e., use $\mathbf{coord}_i = (c_x, c_y, j)$, but not $\mathbf{coord}_i = (c_x, c_y, k)$ described in subsection 3.1. As shown in Table 1a, VecFormer achieves a PQ of 88.4 and 90.9 in the *Total* and *Thing* categories, outperforming the second-best methods by 2.2 and 2.9, respectively. For the more challenging *Stuff* category, VecFormer demonstrates particularly strong performance, achieving a PQ of 85.9 with a notable gain of 21.2 over the second-best result.

These results demonstrate that VecFormer maintains excellent generalization and robustness even without relying on prior information, making it more suitable for practical deployment in real-world CAD scenarios.

**Primitive-Level Semantic Quality.** We assess the model's semantic prediction performance for each graphical primitive by computing the F1 and wF1 score. As summarised in Table 1b, our VecFormer consistently surpasses all prior methods, achieving an improvement of 4.3 in F1 and 3.9 in wF1, compared to SymPoint-V2 [32]. The qualitative results are shown in Figure 3. For more qualitative studies, please refer to Appendix D.

## 4.4 Ablation Studies

**Impact of Sampling Strategy.** As shown in Table 2a, line sampling outperforms point sampling in both settings—with and without prior information. The point sampling variant omits line-specific features $(l, d_x, d_y)$, leading to inferior results, confirming the superiority of line-based representations for vector graphic understanding.

**Choice of Sampling Ratio.** As shown in Table 2b, reducing the sampling ratio $\alpha_{sample}$ from 0.1 to 0.01 steadily improves performance, with the best PQ (91.1) at $\alpha_{sample} = 0.01$—also yielding peak $PQ_{th}$ and $PQ_{st}$ scores. Further reduction to $\alpha_{sample} = 0.005$ slightly degrades performance while

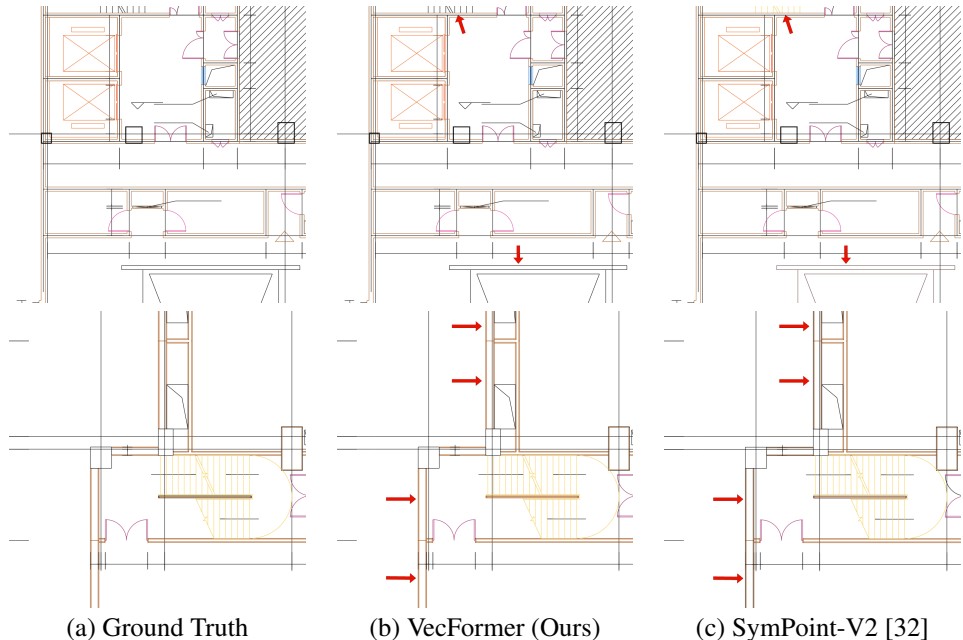

|                 |                    |                    |
|:---------------:|:------------------:|:------------------:|
| (a) Ground Truth | (b) VecFormer (Ours) | (c) SymPoint-V2 [32] |

Figure 3: Qualitative comparison of primitive-level semantic quality between VecFormer and SymPoint-V2. Each row shows a representative example, with (a) Ground Truth annotations, (b) predictions from our VecFormer, and (c) predictions from SymPoint-V2. As shown, VecFormer provides more accurate and consistent semantic predictions across various graphical primitives.

increasing computational cost, making $\alpha_{\text{sample}} = 0.01$ the optimal trade-off between accuracy and efficiency.

Table 2: Ablation studies on sampling strategy, sampling ratio, BFR, and prior information.

(a) Ablation studies on sampling strategy

| Prior | Strategy | PQ | PQ$_{\text{th}}$ | PQ$_{\text{st}}$ |
|:-----:|:--------:|:----:|:----:|:----:|
| w/o | Point | 87.8 | 89.9 | 85.4 |
|     | Line | **88.4** | **90.9** | **85.9** |
| w/ | Point | 89.8 | 90.0 | 89.5 |
|    | Line | **91.1** | **91.8** | **90.4** |

(b) Ablation studies on sampling ratio

| Ratio | PQ | PQ$_{\text{th}}$ | PQ$_{\text{st}}$ |
|:-----:|:----:|:----:|:----:|
| 0.1 | 90.4 | 91.1 | 89.7 |
| 0.05 | 90.6 | 91.3 | 89.7 |
| 0.01 | **91.1** | **91.8** | **90.4** |
| 0.005 | 90.4 | 91.0 | 89.8 |

(c) Ablation studies on BFR

| Method | PQ | PQ$_{\text{th}}$ | PQ$_{\text{st}}$ |
|:------:|:----:|:----:|:----:|
| w/o BFR | 89.2 | 90.4 | 88.1 |
| w/ BFR | **91.1** | **91.8** | **90.4** |
| Gain | (+1.9) | (+1.4) | (+2.3) |

(d) Ablation studies of prior information

| Base | Layer | LFE | PQ | PQ$_{\text{th}}$ | PQ$_{\text{st}}$ |
|:----:|:-----:|:---:|:----:|:----:|:----:|
| ✓ |   |   | 88.4 | 90.9 | 85.9 |
| ✓ | ✓ |   | 90.2 | 91.7 | 88.4 |
| ✓ | ✓ | ✓ | **91.1** | **91.8** | **90.4** |

**Effects of the Branch Fusion Refinement Strategy.** We conduct controlled experiments to evaluate the effectiveness of the proposed Branch Fusion Refinement (BFR) strategy. As shown in Table 2c, incorporating BFR significantly boosts performance across all metrics, demonstrating its essential role in improving prediction accuracy and robustness.

**Effects of Prior Information.** As shown in Table 2d, replacing the primitive ID $j$ with the layer ID $k$ in the position vector boosts PQ from 88.4 to 90.2, highlighting the value of layer priors. Adding the Layer Feature Enhancement (LFE) module further improves PQ to 91.1, demonstrating that structural priors and LFE together enhance geometric understanding.

## 5    Conclusions

We present *VecFormer*, a novel method that employs an expressive and type-agnostic line-based representation to enhance feature learning for vector graphical primitives by preserving geometric continuity and structural relationships, which are critical for symbol-rich vector graphics. To unify instance- and semantic-level predictions from a dual-branch Transformer decoder, we propose the *Branch Fusion Refinement* (BFR) module, which resolves inconsistencies and improves panoptic quality. A current limitation lies in the use of uniform line sampling for simplicity, which may underperform in regions of high geometric complexity. Future work will explore a geometry-aware dynamic sampling strategy to better adapt to diverse structural patterns in vector graphics. To the best of our knowledge, the proposed method does not pose any identifiable negative societal risks.

## Acknowledgements

This work was supported by Shanghai Artificial Intelligence Laboratory, the National Key R&D Program of China (No. 2022ZD0161301), the Shanghai Committee of Science and Technology (No. 22YF1461500), and the National Natural Science Foundation of China (No. 62206046).

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

# Appendix

## A Detailed Visual Comparisons across Different Representations.

We present additional fine-grained visualizations to facilitate a more detailed comparison of different representations. As illustrated in Figure 4, our proposed line-based representation demonstrates closer visual alignment with the ground truth than point-based methods (e.g., SymPoint [8], CADSpotting [9]), effectively preserving geometric continuity and structural integrity across a variety of primitive types.

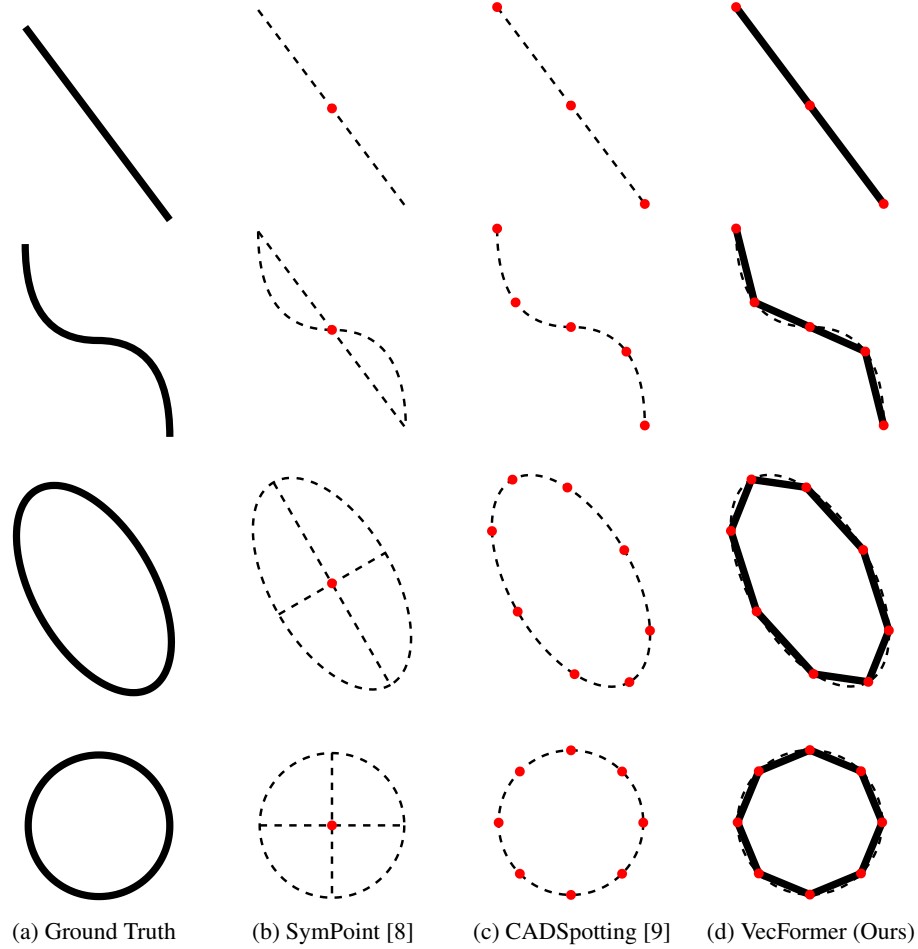

    (a) Ground Truth     (b) SymPoint [8]     (c) CADSpotting [9]     (d) VecFormer (Ours)

Figure 4: Visualization of how different representations perform on different primitives.

# B   Visual Comparison under Varying Sampling Ratios

To further investigate the impact of sampling density on representation quality, we conduct a comparative analysis across different representations under varying sampling ratios $\alpha_{\text{sample}}$. As shown in Figure 5, our line-based representation consistently maintains higher geometric fidelity and structural coherence, even under lower sampling densities. In contrast, point-based representations tend to suffer from fragmentation and loss of continuity as the sampling ratio decreases.

These visual results highlight the robustness of our approach in preserving essential geometric and topological features, suggesting its suitability for vector graphics tasks where structural integrity is critical under constrained sampling conditions.

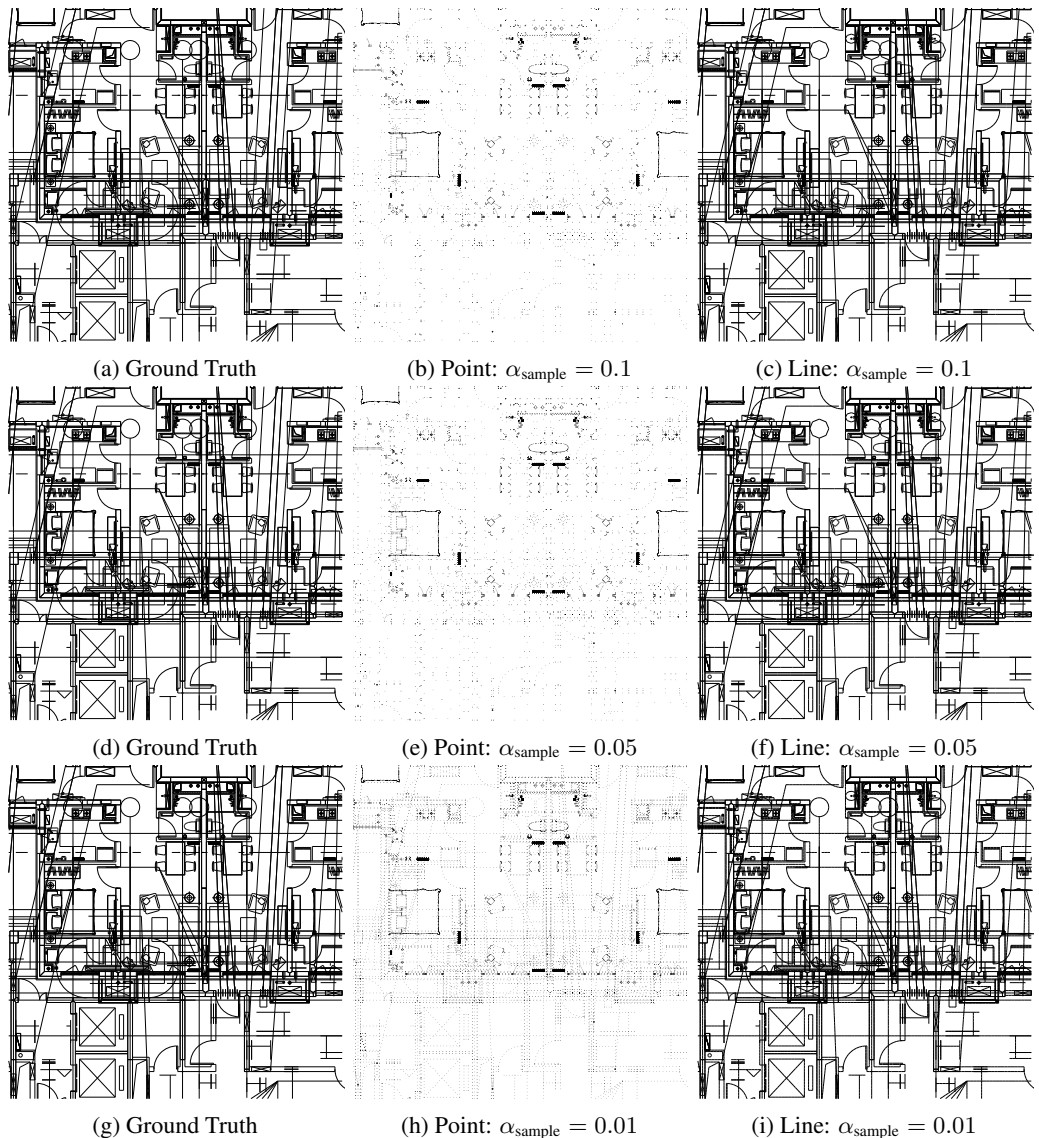

|  |  |  |
|:---:|:---:|:---:|
| (a) Ground Truth | (b) Point: $\alpha_{\text{sample}} = 0.1$ | (c) Line: $\alpha_{\text{sample}} = 0.1$ |
| (d) Ground Truth | (e) Point: $\alpha_{\text{sample}} = 0.05$ | (f) Line: $\alpha_{\text{sample}} = 0.05$ |
| (g) Ground Truth | (h) Point: $\alpha_{\text{sample}} = 0.01$ | (i) Line: $\alpha_{\text{sample}} = 0.01$ |

Figure 5: Visual comparison of the effects of varying sampling ratios on different representations. Since the data is in vector format, zooming in allows for a detailed examination of the differences between representations.

# C   Sequence Length Analysis of Point- and Line-based Representations

This section analyzes the differences in sequence length between point-based and line-based representations on FloorPlanCAD [1] dataset.

We begin by configuring the line-based representation with a sampling ratio of $\alpha_{sample} = 0.01$, consistent with our experimental setup. For the point-based representation, we set $\alpha_{sample} = 0.001$, which yields a similar sampling density to that used in CADSpotting [9], although the sampling strategies differ. As illustrated in Figure 6, this setting results in CADSpotting, the point-based method, producing sequences that are approximately 8 times longer than our line-based counterpart. Despite the significantly shorter sequence length, our method achieves higher Panoptic Quality (PQ), as demonstrated in the main results (Table 1a).

To ensure a fair comparison, we further evaluate both representations under the same sampling ratio of $\alpha_{sample} = 0.01$. Even in this setting, the line-based representation yields approximately 15% fewer tokens than the point-based representation. Moreover, ablation results in Table 2a confirm that our approach not only reduces sequence length but also achieves superior performance.

These findings underscore the efficiency and representational strength of the line-based approach: by encoding primitives through fewer yet structurally meaningful elements, it preserves geometric fidelity while enhancing learning effectiveness. This compact, structure-aware design leads to more accurate segmentation and improved overall performance, making line-based representation a more effective and scalable solution for vector graphic understanding.

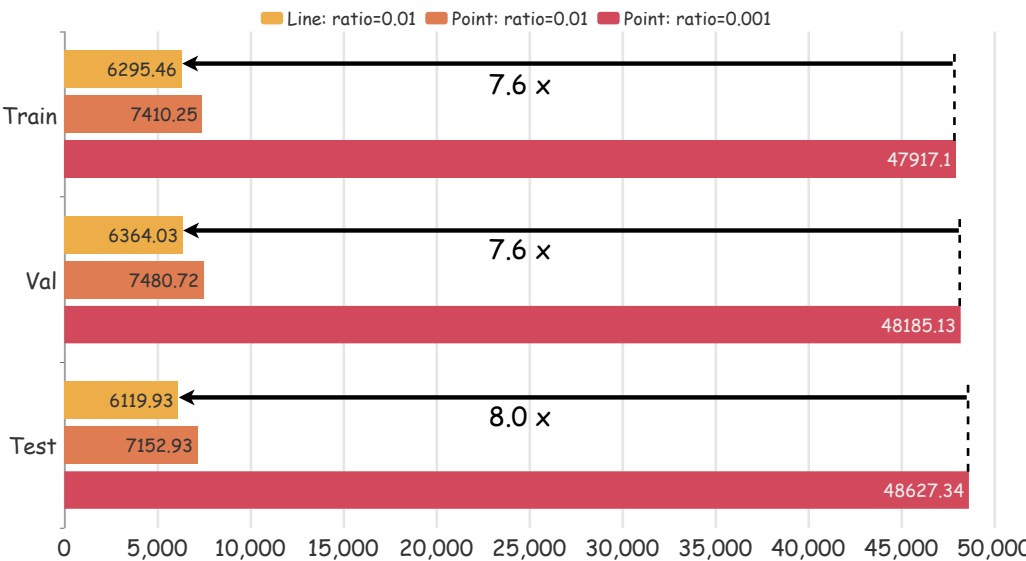

Figure 6: Comparison of sequence lengths between point-based and line-based representations on FloorPlanCAD [1] dataset. The vertical axis indicates different dataset splits, while the horizontal axis represents the average sequence length of each representation across these splits.

# D   Additional Qualitative Evaluation

This section provides additional qualitative results through visualizations. The color scheme for each category is defined in Figure 7, and further examples are illustrated in Figure 8, Figure 9, and Figure 10.

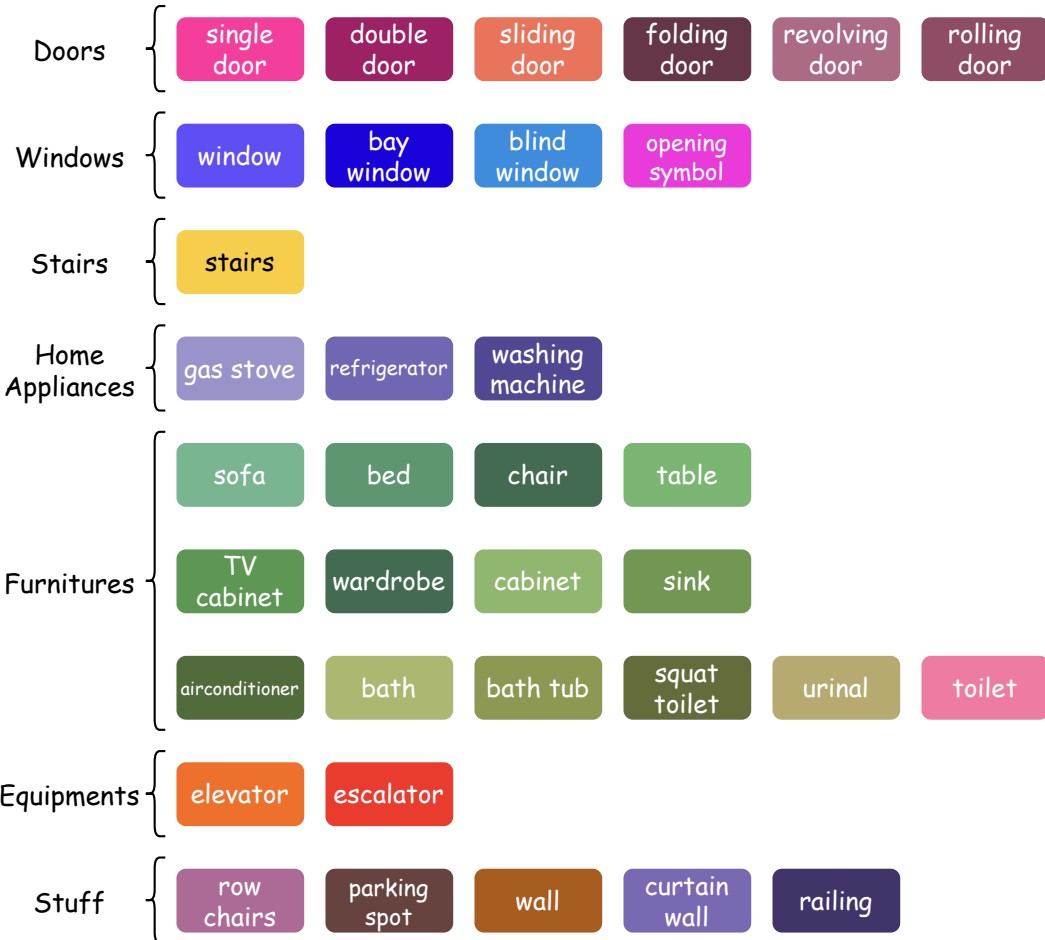

Figure 7: Color map used for category visualization, adapted from [8].

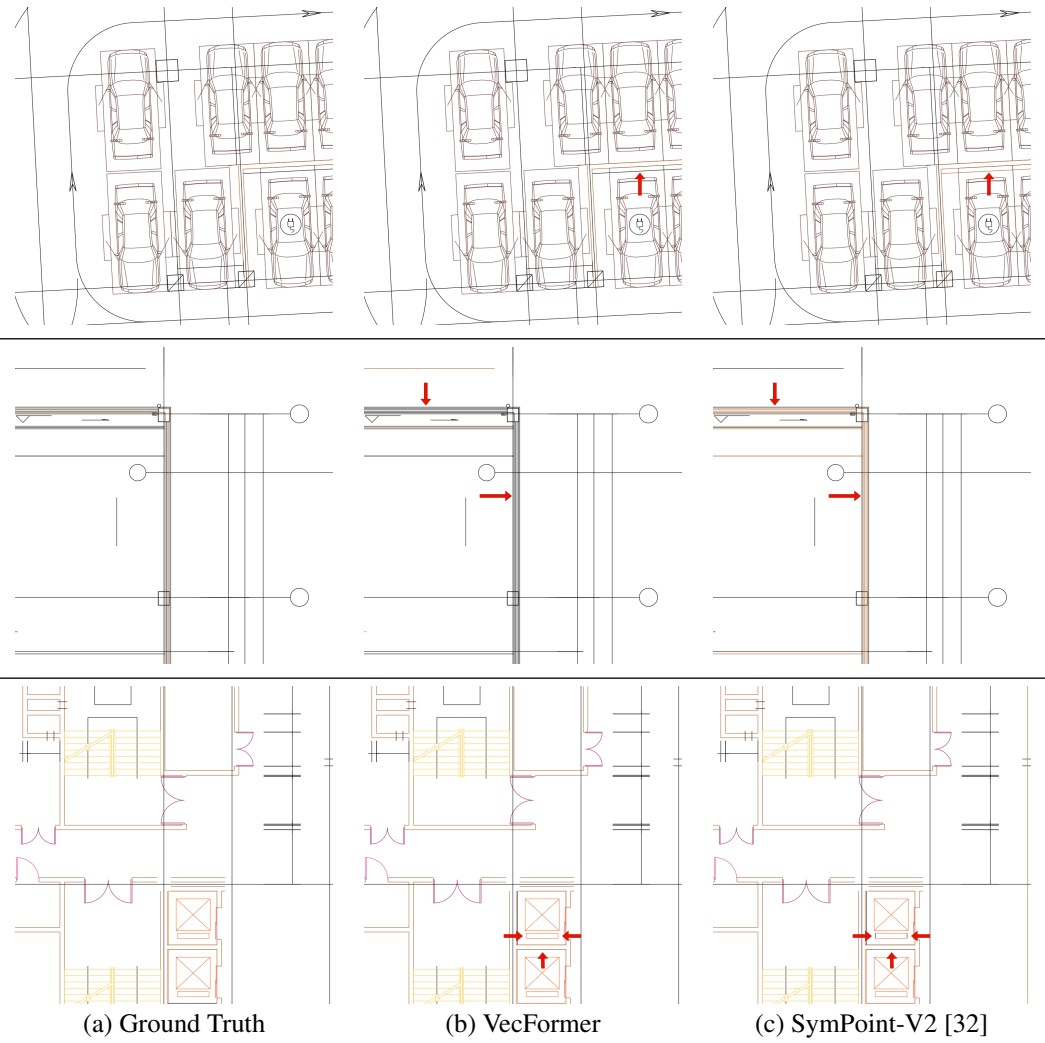

(a) Ground Truth        (b) VecFormer        (c) SymPoint-V2 [32]

Figure 8: More qualitative comparison of primitive-level semantic quality between VecFormer and SymPoint-V2 [32]. Each row shows a representative example, with (a) Ground Truth annotations, (b) predictions from our VecFormer, and (c) predictions from SymPoint-V2 [32]. As shown, VecFormer provides more accurate and consistent semantic predictions across various graphical primitives.

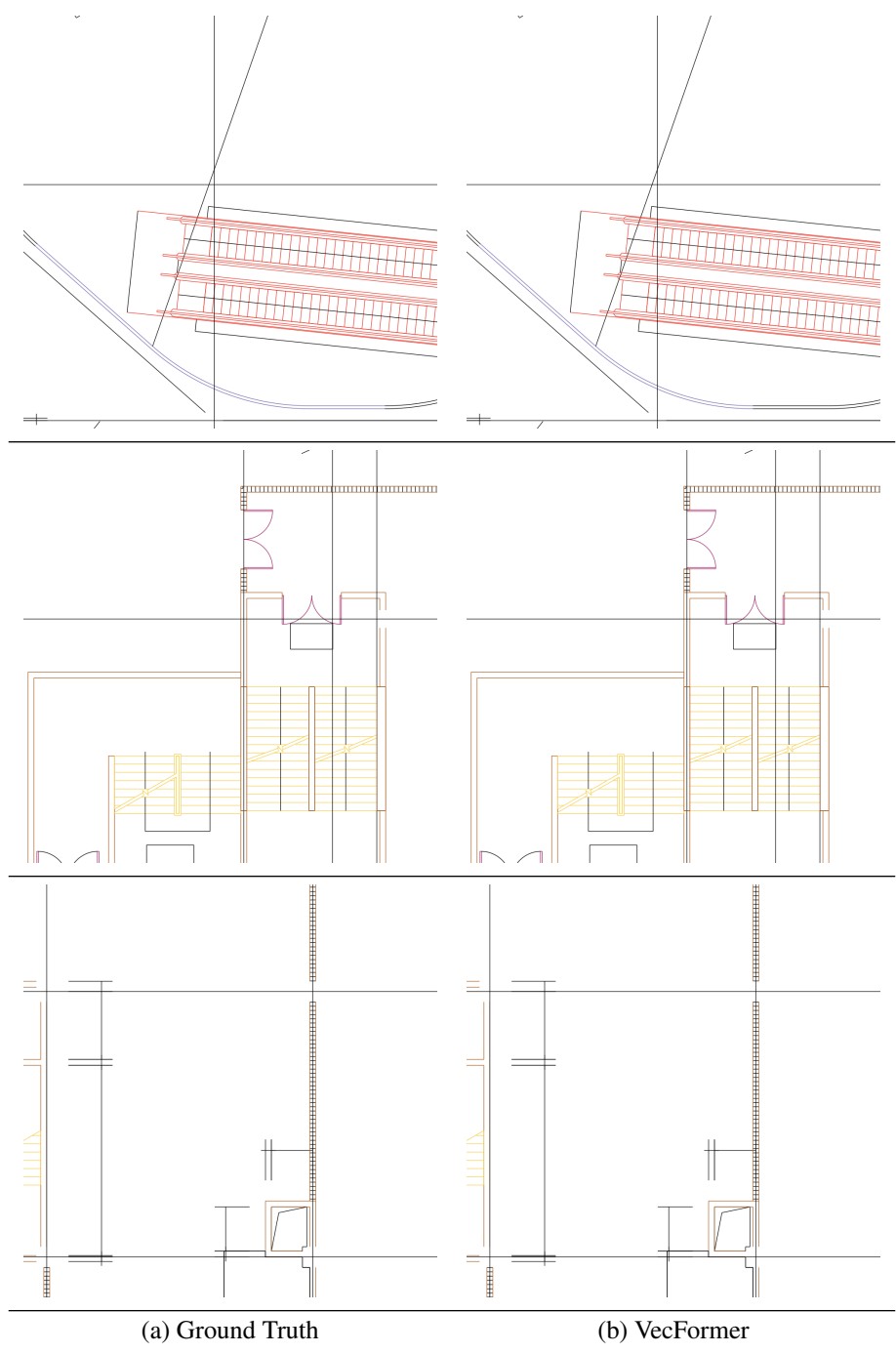

(a) Ground Truth          (b) VecFormer

Figure 9: Results of VecFormer on FloorPlanCAD.

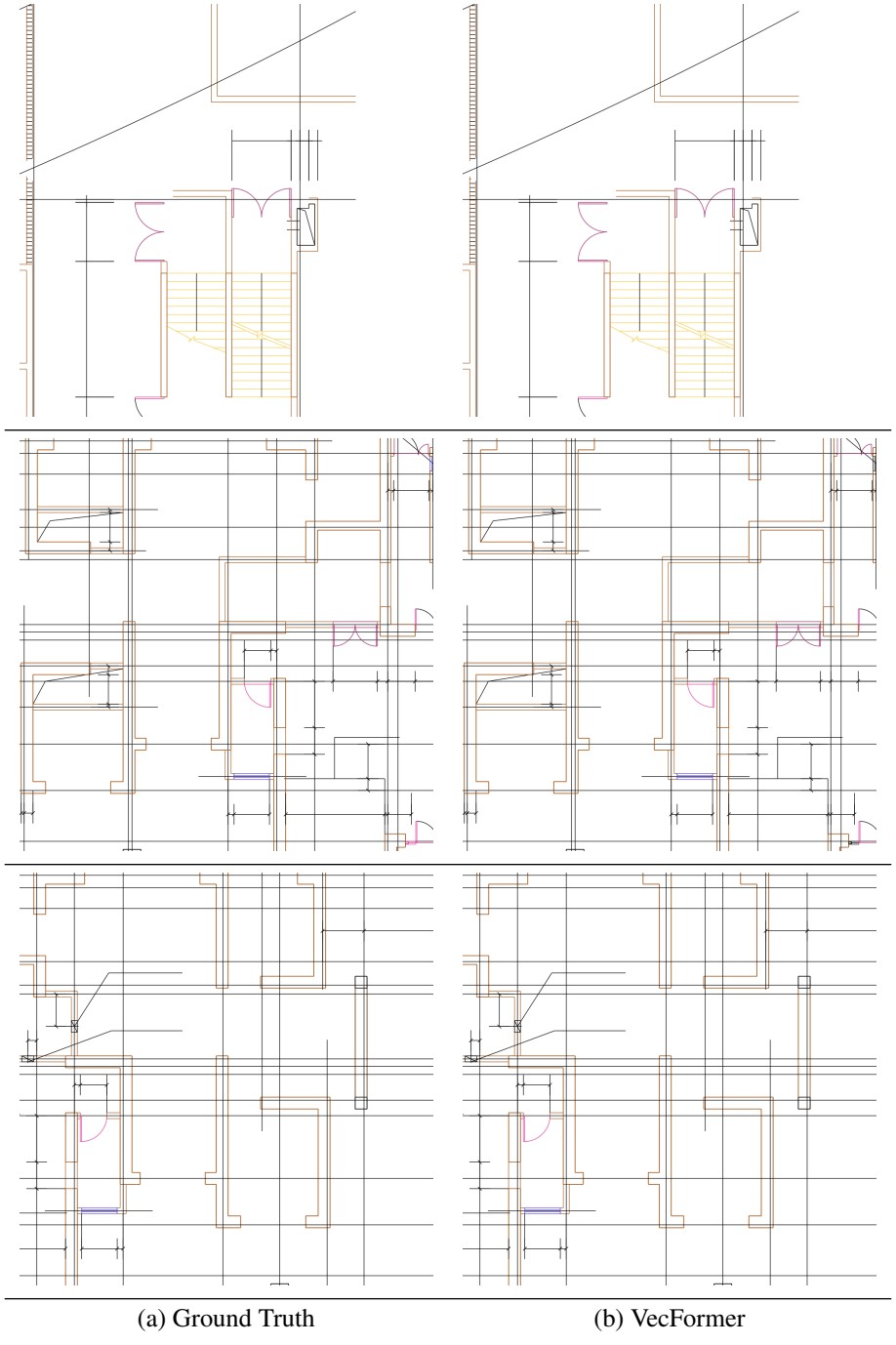

(a) Ground Truth            (b) VecFormer

Figure 10: Results of VecFormer on FloorPlanCAD.

