# OpenReview forum: "Point or Line? Using Line-based Representation for Panoptic Symbol Spotting in CAD Drawings"
_NeurIPS.cc/2025/Conference — NeurIPS 2025 poster_

### Official Review · Reviewer_g5pP · 2025-07-01

**Clarity:** 2
**Significance:** 2
**Originality:** 2
**Rating:** 2
**Confidence:** 3

**Summary:**

The paper is about panoptic symbol spotting, which can be seen as instance/panoptic segmentation on vector graphics. The paper proposed a line-based representation by discretizing vector graphics into a set of line segments. Then an instant/panoptic segmentation method is applied.

**Questions:**

None

**Ethical Concerns:**

["NO or VERY MINOR ethics concerns only"]

**Final Justification:**

Thanks for the response. It solved some of my concerns. Since I am not an expert in this area, I would like to listen other reviewers' opinions.

**Limitations:**

The claim "but these approaches often suffer from high computational costs, limited generality, and loss of geometric structural information" in the abstract is not well-supported. If they are in the abstract which means they are important, they should be emphasized somewhere.

**Quality:**

2

**Strengths And Weaknesses:**

Both the writing and figures are clear to illustrate the core idea. However, I have the following concerns:

1. The method looks like a combination of vector graphics discretization and panoptic segmentation. For the discretization, the authors claim a novel line representation is proposed. However, to me, this is still point based which is similar to prior work. It is just people might be using different pooling methods for the handcrafted features (despite Fig 4 in the appendix).

2. To prove the effectiveness of the proposed representation, I believe the authors need a better experimental setup. For example, the same network backbone (similar number of parameters), the same panoptic segmentation method, ... so that we can ablate the proposed representation does work.

3. I do not know this field very well. However, I checked the competitor methods shown in this paper. They all followed a similar way (see 1). I do not believe the paper has enough novelty. The paper lacks conceptual depth and is quite incremental compared to other similar papers.

4. The large-scale CAD dataset proposed by CADSpotting is missing in the paper. It might be important to test the scaling and generalization ability of the method.

---

> ### Author Rebuttal · Authors · 2025-07-31
>
> Dear Reviewer g5pP,
>
> We sincerely thank you for your time and for providing a thorough and constructive review of our manuscript. We have carefully considered all your comments and will revise our manuscript accordingly.
>
> Below, we address each of your concerns point-by-point.
>
> ## Concern 1: On the novelty of the line-based representation and the support for claims in the abstract.
>
> A core limitation of point-based representations is that they simplify complex vector graphics into discrete points, retaining only intrinsic point attributes such as coordinates. This not only leads to the loss of geometric structure but also requires denser sampling and greater computational overhead to compensate for the missing contextual information.
>
> Our line-based representation adopts a more expressive and structured sampling strategy by sparsely sampling line segments as geometric primitives, instead of using dense point clouds.
>
> This approach enables the explicit encoding of a richer set of intrinsic geometric features (such as length, orientation, and local connectivity, etc) that are inherently lost in point-based sampling. This distinction is not merely an alternative sampling strategy but a shift in the representational paradigm, which we argue is crucial for effectively understanding vector graphics.
>
> To demonstrate its novelty, we will now describe how our line-based representation addresses the three challenges mentioned in the abstract:
>
> - **Addressing** _**Loss of Geometric Structural Information**_**:** Point-based representations reduce primitives to disconnected coordinates, thereby discarding the topological and geometric relationships between them. This approximation inherently leads to a substantial loss of structural information. In contrast, our line-based method explicitly encodes a richer set of intrinsic geometric features (such as length, orientation, and local connectivity, etc) by treating lines as fundamental primitives. Qualitatively, as illustrated in **Figure 1**, our line-based representation visually approximates the original image more closely. Furthermore, as demonstrated in **Appendix B**, our method preserves high visual fidelity even at reduced sampling densities. Quantitatively, the results presented in **Table 2(b) of Section 4.4** show that our method maintains high performance despite a significant reduction in sampling density. These findings collectively demonstrate that the line-based representation more effectively preserves critical geometric information.
>
> - **Addressing** _**High Computational Costs**_**:** For Transformer-based models, the computational cost is proportional to the square of the input sequence length ($O(n^2)$). In **Appendix C**, we analyze the sequence lengths of both representations and find that, at the same sampling density, the sequence length of the line-based representation is approximately 15% shorter than that of the point-based representation. More importantly, as shown in **Appendix B** and **Table 2(b) of Section 4.4**, our line-based representation demonstrates strong robustness to a reduction in sampling density. Even when the sampling density is reduced tenfold, its performance (90.4 PQ) still slightly surpasses the previous state-of-the-art result (90.1 PQ by SymPoint-V2 [1]). This implies that we can achieve or even exceed existing performance with a much shorter sequence length, thereby significantly reducing computational costs. While direct runtime comparisons were outside the scope of this study, this sequence length analysis provides compelling evidence that our method offers a path to significantly lower computational overhead within the same network architecture.
>
> - **Addressing** _**Limited Generality**_**:** Some existing methods, such as SymPoint [2], rely on manually designed sampling and feature engineering rules for different types of primitives. For instance, in the FloorPlanCAD [3] dataset, they designed specific sampling methods for only four primitive types (lines, arcs, circles, and ellipses). This approach lacks generality, as it requires new manual rules when encountering new primitive types (e.g., Bézier curves, complex polylines). In contrast, our method treats all primitives uniformly as vector paths and applies a unified sampling strategy to convert them into a set of line segments. This universal framework allows our method to naturally generalize to vector graphics containing more diverse and complex primitives without any manual modifications. As discussed in **Lines 271-284**, our approach demonstrates exceptional performance even in the absence of priors, which is a critical advantage for real-world applications where such information is often unavailable. Our evaluation shows that existing state-of-the-art methods exhibit a strong reliance on priors, with performance on _Stuff_ categories dropping significantly (e.g., by 31.5 PQ for SymPoint-V2) when priors are removed. In contrast, our method maintains robust performance, achieving a remarkable 21.2 PQ gain over the second-best result in the challenging _Stuff_ category. This highlights the superior practical generality and robustness of our framework, making it more suitable for deployment in real-world scenarios.
>
> In summary, our method is not merely a new representation but an effective solution targeting the core deficiencies of existing approaches. In the revised manuscript, we will restructure the Introduction and Related Work sections to more clearly introduce these arguments and explicitly point to the supporting evidence within the paper (including the appendices) to clarify the motivation and originality of our work.
>
> ## Concern 2: The need for a better experimental setup for ablation.
>
> We thank you for this valuable suggestion. In fact, we had already conducted such an experiment in our original manuscript (**Line 291-294** and **Table 2(a) of Section 4.4**).
>
> We performed a direct comparative ablation study between the line-based and point-based representations.
>
> - We employed the **exact same network architecture and hyperparameter settings** to process both the point-based and line-based representations.
>
> - For the point-based baseline, we simply **removed the higher-order geometric features unique to lines** (e.g., length and orientation) from the input feature vector, leaving only coordinate information. All other aspects of the model remained identical to our full model.
>
>
> The results of this experiment are presented in the original manuscript in **Table 2(a) of Section 4.4** (discussed in **Line 291-294**). The table clearly shows that our line-based representation significantly outperforms the point-based one, both with and without the use of prior information. This provides strong evidence that our proposed representation captures geometric information more effectively, thereby boosting model performance. We will emphasize the setup and conclusion of this experiment more prominently in the revised version to make it more explicit.
>
> ## Concern 3: Missing evaluation on the large-scale CAD dataset.
>
> We thank you for this suggestion. We also believe that evaluating our method on a large-scale dataset like LS-CAD is important for demonstrating its scalability and generalization. However, to the best of our knowledge, the LS-CAD dataset mentioned in the CADSpotting [4] paper has not yet been publicly released. Therefore, we are currently unable to conduct experiments on it.
>
> Should the LS-CAD dataset become publicly available, we are committed to evaluating our method on it and will report the analysis in future work. We will add a note to this effect in the conclusion or future work section of our revised manuscript.
>
> ## References
>
> [1] Wenlong Liu, et al. "Sympoint Revolutionized: Boosting Panoptic Symbol Spotting with Layer Feature Enhancement." _arXiv preprint arXiv:2407.01928_ (2024).
>
> [2] Wenlong Liu, et al. "Symbol as Points: Panoptic Symbol Spotting via Point-based Representation." _arXiv preprint arXiv:2401.10556_ (2024).
>
> [3] Zhiwen Fan, et al. "FloorPlanCAD: A Large-Scale CAD Drawing Dataset for Panoptic Symbol Spotting." _arXiv preprint arXiv:2105.07147_ (2021).
>
> [4] Jiazuo Mu, et al. "CADSpotting: Robust Panoptic Symbol Spotting on Large-Scale CAD Drawings." _arXiv preprint arXiv:2412.07377_ (2024).

---

### Official Review · Reviewer_BWhE · 2025-07-01

**Clarity:** 1
**Significance:** 2
**Originality:** 2
**Rating:** 3
**Confidence:** 3

**Summary:**

The paper tackles the problem of classification and counting of vector graphic CAD drawings to classify and enumerate (or group) the curve primitives within. The work uses a transformer architecture as well as other enchancements, e.g., LFE and BFR, to achieve state-of-the-art performance on the FloorPlanCAD dataset. The contribution is primarily just this: the new architecture and improved performance for this specific task. The tools used for the method, e.g., line-based features and the transformer architecture seem to be relatively natural generalizations and applications of existing methods.

**Questions:**

These are mostly clarity questions, that I think could be easily addressed with some small bits of text.

1. Line 128: Please clarify the types of primitives. Is a "line" primitive necessarily a straight line segment, or could it be a Bezier curve? Later on "line" is used for line segments sampled from primitives, so I wasn't sure. Are there other types of primitives, e.g., fill regions or polygons?
2. Line 141: Where is PTv3 used? The section outlines features that are explicitly given by formulae and then pooling. Are these then fed into PTv3? This seems to conflict with the diagram in Figure 2, where the transformer backbone precedes the pooling of line features.
3. Line 155: What are the layer IDs? Does each one relate to the semantic label? Why does it need to be normalized?
4. Eq (5): Is there a reason you explicitly have the first 5 entries instead of just putting the 2 endpoints of the line segment? They could be derived from just these 2, and perhaps it would lead to a more compact model (or not, who knows). Maybe an ablation would be interesting.
5. Section 3.2.3: Could this section be expanded to be a bit more explicit, at the cost of compressing 3.2.1? The formulae in 3.2.1 are relatively obvious and could be relegated to an appendix, in my opinion.

**Ethical Concerns:**

["NO or VERY MINOR ethics concerns only"]

**Limitations:**

Yes.

**Paper Formatting Concerns:**

None that I saw.

**Quality:**

2

**Strengths And Weaknesses:**

Strengths:
1. The paper provides state-of-the-art performance on a relevant dataset.
2. A novel architecture and set of features for the specific problem are used.
3. Competing methods have been published at ICCV and other top-tier conferences, so while the problem seems a bit niche, it has warranted publication in such venues.

Weaknesses:
1. It will be hard to reproduce the method without a link to code or a repo.
2. I found the presentation of the problem at hand to be unclear. It took me until the Experimental Results section to understand what semantic and instance labels would be. There are also other clarity issues with respect to basic definitions for readers unfamiliar with the Panoptic Symbol Spotting problem. See "Questions" below.
3. I am not an expert in deep learning architectures and their application in this problem domain, but it's unclear to me if insights in this work will extend to other problems.

---

> ### Author Rebuttal · Authors · 2025-07-31
>
> Dear Reviewer BWhE,
>
> We are very grateful for your detailed and insightful review. Your constructive comments have been instrumental in helping us enhance the clarity and impact of our work. We are pleased that you recognized our method's state-of-the-art performance and the significance of the research problem.
>
> Below, we address each of your concerns.
>
> ## Concern 1: On ensuring the reproducibility of our work.
>
> We agree that reproducibility is crucial for scientific advancement. To ensure our results can be fully and easily reproduced, we commit to making our complete project public upon the paper's acceptance. This release will include the full source code for both training and inference, along with the final model weights.
>
> ## Concern 2: On the clarity of the problem definition and key terminology.
>
> Thank you for highlighting this critical point. We agree that a clear, upfront definition is essential for the paper's accessibility. We will significantly revise the Introduction to explicitly define the **Panoptic Symbol Spotting** task, its objectives, and all key terminology at the very beginning. The revised section will ensure that readers not deeply familiar with this specific field can grasp the core challenges and our contributions without needing to read ahead.
>
> ## Concern 3: On the generalizability of our method beyond architectural CAD drawings.
>
> We appreciate you raising this important point. While our method is demonstrated on architectural CAD drawings, we believe the core architectural concepts are applicable to other domains that utilize **structured vector graphics**. The central idea of using a line-based representation to approximate vector graphics is fundamentally not limited to architectural drawings. This approach could be extended to other technical illustration domains, such as recognizing components in electrical schematics or mechanical engineering diagrams, where symbols are defined by arrangements of geometric primitives. We will add a brief discussion on these potential future directions to the Conclusion section.
>
> ## Concern 4: On clarifications regarding methodology and paper structure.
>
> Thank you for the detailed questions, which help us identify areas for improvement. We address them below in order:
>
> 1. **Line 128 (Clarification of "Line" Primitives):** Thank you for pointing out this ambiguity. In this paper, we consistently use the term '**line**' as a shorthand for a '**straight line segment**'. A 'line' primitive can serve as its own sampled representation (a single line segment) without any loss of geometric fidelity. In contrast, other primitives, such as curves, must be discretized into multiple segments to approximate their geometry. This fundamental distinction warrants the application of a different initial sampling number (K) for 'line' primitives compared to other types.
>
> 2. **Line 141 (Use of PTv3):** The **Transformer Backbone** in our framework is indeed **PTv3** [1]. The data pipeline is as follows: 1) The **Line Sampling** module generates a set of line segments. 2) These segments' endpoint coordinates are encoded into position and feature vectors (as detailed in **Section 3.2.1**). 3) These vectors are fed directly into **PTv3**, which processes them to produce contextualized features. 4) The **Line Pooling** module operates on these output features. We used the general term _Transformer Backbone_ in Figure 2 to highlight the modularity of our framework; while we selected PTv3 for our implementation, it can be substituted with other similar backbone architectures.
>
> 3. **Line 155 (Layer IDs):** In CAD drawings, layers are used to group related entities. For example, a layer named "doors" might contain various types of doors (single, double, sliding). By including the layer ID, we provide the model with this crucial semantic grouping information. We **normalize** this information to ensure that all dimensions of the input position vector are on a uniform scale (i.e., within the ($[-0.5, 0.5]$) range). This prevents any single feature from dominating the distance calculations and improves the numerical stability of the model during training. We will expand this explanation in the paper.
>
> 4. **Eq (5) (Feature Formulation):** This is an excellent observation that touches upon a key aspect of our model's design. While these features are geometrically derivable, we found empirically that providing them explicitly leads to a significant performance improvement. Actually, our earliest models, which used only endpoint coordinates, underperformed compared to the current feature set. Our hypothesis is that while these higher-level geometric features are derivable from the endpoints, it is non-trivial for the model to learn these relationships implicitly. By providing higher-level features explicitly, we allow the model to more effectively leverage the key advantage of a line-based representation: the rich geometric information that is inherently lost in point-based methods. Below are the results of the ablation study:
>
> |              |             |             |             |
> | :----------: | :---------: | :---------: | :---------: |
> |Features|PQ|PQ$_{\text{th}}$|PQ$_{\text{st}}$|
> | 2 endpoints  |    89.2     |    90.7     |    87.7     |
> | **Higher-level** | **91.1**${\scriptsize(+1.9)}$ | **91.8**${\scriptsize(+1.1)}$ | **90.4**${\scriptsize(+2.7)}$ |
>
> 5. **Section 3.2.3 vs. 3.2.1:** Thank you for this constructive suggestion. We agree that it would improve the paper's flow and clarity. In the revised manuscript, we will condense **Section 3.2.1** by moving the detailed formulas to an appendix. We will then use the freed space to provide a more intuitive and thorough explanation of the LFE module's functionality in **Section 3.2.3**.
>
> We are confident that these revisions will significantly improve the quality and clarity of our paper. Thank you once again for your time and insightful feedback.
>
> ## References
>
> [1] Xiaoyang Wu, et al. "Point Transformer V3: Simpler, Faster, Stronger." _arXiv preprint arXiv:2312.10035_ (2023).

---

### Official Review · Reviewer_hNNU · 2025-07-03

**Clarity:** 3
**Significance:** 3
**Originality:** 3
**Rating:** 5
**Confidence:** 4

**Summary:**

The paper targets panoptic symbol spotting task which is a challenging task in the domain of vector drawing analysis aiming to predict semantic and instant labels for countable and amorphous symbols in the input vector drawing. To address this task, the authors propose to enrich the input representation available to the deep learning approaches by approximating all input primitives by sequences of straight lines. This enables a more complete input representation that pairs well with the modern deep learning architectures such as transformers. The authors then use this input representation to feed into a point transformer model and refined using a conventional refinement strategy to consolidate semantic and instance predictions and confidence scores. A thorough evaluation is performed on the FloorPlanCAD dataset that shows how the proposed approach compares both regarding the baselines and regarding various design choices.

**Questions:**

The only concern I have right now regarding the experimental evaluation is regarding the use of the BFR strategy used to refine results produced by the raw method. I think that the most compelling way to rebut that would be to conduct an experiment with one of the baseline methods and basically refine its results also using the BFR strategy. I do not know if this is an easy task or if it makes total sense to executive this comparison and exactly this way.

**Ethical Concerns:**

["NO or VERY MINOR ethics concerns only"]

**Final Justification:**

I have read the reviews and the authors' replies to them. I'm inclined to keep my original ratings, and I hope those were encouraging to the authors.

I do not know if the other reviewers shall see this note, but it seems to me that the authors had in mind a practical, concrete task, for which they developed and studied a concrete solution. My experience with vectorization tasks lays exactly within the architectural/mechanical drawings domains, and an accurate symbol segmentation would definitely not harm the final result for those kinds of tasks. I would imaging that a solid segmentation would be quite useful for retrieval, too.

So while I can agree that the authors might have sticked to a bit safer plan in terms of technical methodology (obtaining perhaps less "exciting" results), I still believe their work constitutes visible progress on a useful image processing problem. Thus I'm in favor of this paper being accepted.

**Limitations:**

Yes

**Paper Formatting Concerns:**

No concerns

**Quality:**

3

**Strengths And Weaknesses:**

Strengths:
1. The effectiveness of the proposed input representation is clear and compelling. I also appreciate that the authors provides the districts on the number of samples required to represent and given input drawing with various representations and how their method is more economical compared to the baseline.
2. The effectiveness of the series of design stages in the proposed approach is demonstrated in the compelling way.
3. The experimental validation is reasonable compelling and shows the advantages of the proposed approach quite clearly.
4. The paper is sufficiently verbose with respect to the design of the method the experiments conducted and the outcomes. I believe that all the necessary information to reproduce the findings of this paper is presented.

Weaknesses:
1. The only major weakness that I see at this point is the performance drop from not including the Branch Fusion Refinement (BFR) strategy. This begs the question whether including branch fusion refinement in the baseline approaches would also improve their performance?

---

> ### Author Rebuttal · Authors · 2025-07-31
>
> Dear Reviewer hNNU,
>
> Thank you for your positive and insightful feedback on our work. We are very encouraged by your appreciation for our proposed line-based representation, the rigor of our experimental validation, and the overall clarity of our paper. We are especially grateful for your constructive suggestion regarding our Branch Fusion Refinement (BFR) strategy.
>
> ## Regarding your question on the Branch Fusion Refinement (BFR) strategy:
>
> Your question regarding the generalizability of our BFR strategy and whether it would confer similar performance enhancements to baseline methods is an excellent one. We appreciate this critical question, as it prompted us to conduct further experiments to address this matter directly.
>
> You correctly observed that the performance degradation without BFR highlights its importance within our framework. However, the BFR module, as its name (Branch Fusion Refinement) suggests, is intrinsically designed to fuse predictions from a multi-branch decoder architecture. To the best of our knowledge, most existing baseline methods for this task employ a single-branch decoder, which renders them architecturally incompatible with our BFR module.
>
> To empirically validate this and directly address your concern, we identified a baseline with a compatible multi-branch structure, **CADSpotting** [1]. We then integrated our BFR strategy into the CADSpotting model to evaluate its impact. The results are presented below:
>
> |                   |                           |                           |                           |
> | :---------------: | :-----------------------: | :-----------------------: | :-----------------------: |
> |      Method       |            PQ             |     PQ$_{\text{th}}$      |     PQ$_{\text{st}}$      |
> |    CADSpotting    |           88.9            |           88.7            |           81.8            |
> | CADSpotting + BFR | 89.9${\scriptsize(+1.0)}$ | 89.8${\scriptsize(+1.1)}$ | 84.7${\scriptsize(+2.9)}$ |
> |     **VecFormer(Ours)**      |         **91.1**          |         **91.8**          |         **90.4**          |
>
> As the results indicate, applying our BFR strategy to CADSpotting yields a consistent and notable performance improvement across all metrics. This outcome confirms your intuition and demonstrates that our BFR module is indeed a generalizable component that can benefit other multi-branch architectures.
>
> Crucially, however, even with the BFR enhancement, the performance of CADSpotting remains significantly below that of our proposed method. We believe this comparison strongly substantiates our central claim: the superior performance of our method is primarily driven by our **novel line-based representation**. This distinction is crucial, as it confirms that our core contribution, the representation itself, is the principal driver of our results, not just the refinement module. We believe this new experiment robustly addresses your concern and further validates the primary contributions of our paper.
>
> This line of inquiry, prompted by your valuable suggestion, has also inspired a new direction for our future work: exploring the application of BFR to other domains that utilize multi-branch models, such as image segmentation. This will allow us to further test and hopefully confirm the broader utility of the BFR strategy.
>
> Thank you once again for your time and your constructive review.
>
> ## References
>
> [1] Jiazuo Mu, et al. "CADSpotting: Robust Panoptic Symbol Spotting on Large-Scale CAD Drawings." _arXiv preprint arXiv:2412.07377_ (2024).

---

### Official Review · Reviewer_Ye7H · 2025-07-03

**Clarity:** 3
**Significance:** 3
**Originality:** 4
**Rating:** 5
**Confidence:** 4

**Summary:**

This paper presents VecFormer, a novel method for panoptic symbol spotting in CAD drawings by introducing a line-based representation instead of conventional point-based or rasterized representations. The key idea is to preserve geometric continuity and structural integrity by approximating primitives with sequences of line segments. The authors design a Transformer-based framework that leverages this representation, combined with a dual-branch decoder for instance and semantic prediction and a lightweight Branch Fusion Refinement module to improve label coherence. Extensive experiments on the FloorPlanCAD dataset demonstrate that VecFormer achieves new state-of-the-art results in terms of Panoptic Qualit and Stuff-PQ, with clear improvements over existing methods such as SymPoint-V2 and CADSpotting.

**Questions:**

(1) On broader generalization: Have the authors considered validating VecFormer on other vector graphic tasks or datasets to strengthen the generalizability claim? Even small-scale tests (e.g., technical drawings, maps) would help.
(2) Sampling strategy: Could the authors provide empirical evidence comparing uniform sampling with an adaptive or curvature-aware sampling strategy? This is important given the stated limitation.
(3) End-to-end refinement: Is it possible to integrate the Branch Fusion Refinement module as a learnable layer within the model instead of using it purely as post-processing? A brief discussion or experiment would improve the contribution.
(4) Reproducibility: Will the authors release the complete training and inference code along with pretrained models? This is essential for verifying the impressive gains.
(5) Statistical robustness: Could the authors provide standard deviations or error bars for main metrics over multiple seeds to support the robustness of the results?

**Ethical Concerns:**

["NO or VERY MINOR ethics concerns only"]

**Final Justification:**

Many thanks to authors for their detailed responses. They raised various good points, although I keep my score the same, as it was on the positive side.

**Limitations:**

Yes, the paper discusses a key limitation of uniform line sampling and suggests future directions to address it. However, the impact of this limitation is not empirically shown. I encourage the authors to either visualize failure cases or provide additional experiments to clarify when uniform sampling might fail.

**Paper Formatting Concerns:**

There are no obvious formatting issues.

**Quality:**

3

**Strengths And Weaknesses:**

Strengths

Originality: The proposed line-based representation is conceptually simple yet effective, addressing key limitations of point-based methods.

Technical Quality: The paper provides solid methodological details, including clear mathematical formulations, ablation studies, and implementation settings. The BFR module is a thoughtful addition to improve panoptic consistency.

Significance: The improvement on Stuff-PQ is particularly noteworthy, given that this aspect is challenging for prior methods. The approach is well motivated for practical CAD applications.

Clarity: The paper is generally clear, with good figures and detailed appendices for qualitative results and additional analyses.

Weaknesses

While the line-based representation is well motivated, its potential limitations are only briefly mentioned and not empirically examined.

The experiments focus solely on the FloorPlanCAD benchmark. The generalizability of VecFormer to other vector graphic domains is claimed but not tested.

The paper does not provide statistical significance analysis for the reported results.

The BFR refinement is post-processing but could potentially be integrated more tightly within the model to allow end-to-end training, which is not discussed.

---

> ### Author Rebuttal · Authors · 2025-07-31
>
> Dear Reviewer Ye7H,
>
> We sincerely thank you for your positive and constructive review. We are encouraged by your assessment of our proposed line-based representation as original and effective, and our method as technically solid. Your insightful comments have been invaluable for improving our manuscript, and we will address each of your questions below.
>
> ## Concern 1: On broader generalization
>
> We thank you for this important question and agree that demonstrating our method's generalizability is crucial.
>
> To our knowledge, **FloorPlanCAD** was the only publicly available dataset for the Panoptic Symbol Spotting task. To further validate our approach, we actively sought additional data. We identified a recent work, **ArchCAD-400k** [1], which introduced a larger and more complex dataset for this task. Although the full dataset is not yet public, we contacted the authors and were fortunate to obtain a 40k subset. This subset alone is significantly larger and more richly annotated than the entire FloorPlanCAD dataset (which contains approximately 10k samples).
>
> We evaluated our method on this new, more challenging data and compared it against the existing state-of-the-art method, **SymPoint-V2** [2]. The results are presented below:
>
> |                      |                                |                               |                                |
> | :------------------: | :----------------------------: | :---------------------------: | :----------------------------: |
> |        Method        |               PQ               |       PQ$_{\text{th}}$        |        PQ$_{\text{st}}$        |
> |     SymPoint-V2      |              62.4              |             64.9              |              53.7              |
> | **VecFormer (Ours)** | **72.6**${\scriptsize(+10.2)}$ | **70.1**${\scriptsize(+5.2)}$ | **79.4**${\scriptsize(+25.7)}$ |
>
> As the table shows, our method maintains a significant performance advantage, providing strong evidence for its generalization capabilities. Regarding other vector graphic domains, we have not yet found other publicly available datasets with the necessary panoptic annotations for a direct comparison.
>
> We will add these new findings and a detailed description of the dataset subset to our revised manuscript. We are also committed to evaluating VecFormer on other relevant datasets as they become available.
>
> ## Concern 2: On the sampling strategy
>
> Thank you for this insightful question regarding our sampling strategy.
>
> Our choice of uniform sampling was initially motivated by the geometric properties of the FloorPlanCAD dataset. The vast majority of primitives in this benchmark are either straight line segments, which are perfectly represented by our sampling, or circular arcs with smooth, low-frequency curvature. For these primitives, uniform sampling provides a simple yet effective means of generating accurate line-based approximations.
>
> However, we concur that for more complex curves with high-frequency variations, such as the intricate Bézier curves found in graphic design, an **adaptive, curvature-aware sampling strategy** would likely yield superior performance. Intuitively, such a strategy would allocate more sample points to regions of high curvature to ensure a faithful approximation, while using fewer points in flatter regions. This would not only improve accuracy but also optimize the sequence length, mitigating the redundancy that can arise when a high uniform sampling rate is applied globally.
>
> We acknowledge that these are currently intuitive hypotheses that we have not yet validated with empirical experiments. We believe this is a very promising direction for future work and will add a discussion of this limitation, along with the potential of adaptive sampling, to our paper.
>
> ## Concern 3: On end-to-end refinement
>
> This is an excellent suggestion. The primary motivation for designing the Branch Fusion Refinement (BFR) module as a post-processing step was its simplicity, efficiency, and modularity. It is a lightweight module that can be easily integrated to refine the outputs of any two-branch panoptic spotting framework without requiring end-to-end retraining.
>
> Integrating the BFR module as a learnable layer for end-to-end training is indeed a fascinating prospect. Such a module could potentially learn more complex and powerful data-driven strategies for fusing semantic and instance predictions. However, this direction also presents non-trivial challenges. For instance, in our current architecture, the semantic and instance branches are guided by two distinct supervision signals. An end-to-end approach would require a carefully designed architecture to effectively fuse these disparate signals, which could introduce significant complexity and potential instability to the training pipeline.
>
> We agree this is a valuable avenue for future research. We will add a discussion to the limitations section of our revised manuscript, elaborating on our current design choice and the potential, as well as the challenges, of an end-to-end BFR module.
>
> ## Concern 4: On reproducibility
>
> Yes. We are fully committed to open and reproducible research. Upon acceptance, we will release the complete source code for training and inference, all pretrained models, and detailed instructions to reproduce our results.
>
> ## Concern 5: On statistical robustness
>
> Thank you for raising this important point. As retraining the model multiple times incurs significant computational costs, we instead conducted multiple inference experiments on the test set using different random seeds. The results are as follows:
>
> |                  |                              |                              |                              |
> | :--------------: | :--------------------------: | :--------------------------: | :--------------------------: |
> |      Method      |              PQ              |       PQ$_{\text{th}}$       |       PQ$_{\text{st}}$       |
> | VecFormer (Ours) | 91.1${\scriptsize(\pm 0.1)}$ | 91.9${\scriptsize(\pm 0.1)}$ | 90.3${\scriptsize(\pm 0.2)}$ |
>
> Once again, we thank you for your valuable feedback, which has helped us significantly strengthen our manuscript.
>
> ## References
>
> [1] Ruifeng Luo, et al. "ArchCAD-400k: An Open Large-Scale Architectural CAD Dataset and New Baseline for Panoptic Symbol Spotting." _arXiv preprint arXiv:2503.22346_ (2025).
>
> [2] Wenlong Liu, et al. "Sympoint Revolutionized: Boosting Panoptic Symbol Spotting with Layer Feature Enhancement." _arXiv preprint arXiv:2407.01928_ (2024).

---

### Decision · Program_Chairs · 2025-09-17

**Decision:**

Accept (poster)

**Comment:**

This paper proposes VecFormer, introducing a line-based representation for panoptic symbol spotting in CAD drawings, combined with a dual-branch architecture and a Branch Fusion Refinement module. The method is well motivated, addressing limitations of point-based and rasterized approaches by preserving geometric continuity and enabling more efficient representations. Experiments show strong improvements, particularly in Stuff-PQ, establishing new state-of-the-art performance on FloorPlanCAD and demonstrating generalization to additional data. While one reviewer raised concerns about novelty, they explicitly noted a lack of expertise in this area, and the other reviews—both confident and positive—found the contribution original, technically solid, and practically relevant. The area chair concurs with the positive assessments and views the method as clear progress on an important and established research problem. I recommend acceptance.